# HIRA defines early replication initiation zones independently of their genome compartment

Tina Karagyozova[1,6], Alberto Gatto [1], Audrey Forest [1], Jean-Pierre Quivy [1], Rocío Nunez-Vazquez [1], Marc A. Martí-Renom[2,3,4], Leonid A. Mirny [5] & Geneviève Almouzni [1] ✉

Chromatin states and 3D architecture have been used as proxy to identify replication initiation zones (IZs) in mammalian cells, yet their functional interconnections remain a puzzle. Here, to dissect these relationships, we focus on the histone H3.3 chaperone HIRA recently implicated in early initiation zone (IZ) definition. We monitor 3D organisation, chromatin accessibility and histone post-translational modifications (PTMs) in wild-type and HIRA knock-out cells in parallel with early replication initiation. In the absence of HIRA, compartment A loses H3.3 enrichment and gains accessibility without changes in associated histone post-translational modifications (PTMs). Furthermore, impaired early firing at HIRA-dependent IZs does not correspond to changes in chromatin accessibility or patterns of histone H3 PTMs. Additionally, a small subset of early IZs initially in compartment A switch to B and lose early initiation in the absence of HIRA. Critically, HIRA complementation restores these early IZ, and H3.3 variant enrichment, without substantial compartment reversal. Thus, while HIRA contributes to compartment A features, its role in regulating early replication initiation can be uncoupled from accessibility, histone marks and compartment organisation.

The synthesis of new DNA starts at the origins of replication: estimated 30,000–50,000 sites in the mammalian genome. These origins are licensed in G1 but only a small subset are used stochastically in S phase to initiate DNA synthesis[1]. In metazoans, origin definition combines properties of DNA sequence[2,3] and epigenetic features[4–6]. Initiation events are typically clustered in initiation zones (IZs) of 20–150 kb[2,4,7–9], which do not fire at the same time, but follow a spatiotemporal order of activation, termed the replication timing (RT) programme[10]. RT is cell-type specific and strongly correlates with chromatin state and its three-dimensional (3D) organization[10]. Typically, early-replicating regions are transcribed[3], accessible[11], decorated by active histone marks[12] and correspond to compartment A identified by Hi-C[13–15]. Few exceptions to these correlations have been identified in early development[16–18], but RT and chromatin features are not fully established at this stage, making it challenging to translate this in the context of differentiated cells. Thus, a major question in the field has been to disentangle the functional connections between early replication, epigenetic state, and 3D chromatin organization.

Recently, a role for histone variants has also emerged in the context of the regulation of replication initiation[19–21]. With respect to histone H3, we first found that enrichment pattern of H3.3 and H3.1 followed early versus late RT respectively[22]. The replicative H3.1 variant,

[1]Institut Curie, PSL Research University, Sorbonne Université, CNRS UMR3664, Laboratoire Dynamique du Noyau, Equipe Labellisée Ligue contre le Cancer, Paris, France. [2]Centre Nacional d'Anàlisi Genòmica (CNAG), Barcelona, Spain. [3]Centre for Genomic Regulation (CRG), Barcelona Institute of Science and Technology (BIST), Barcelona, Spain. [4]ICREA, Pg. Lluís Companys 23, Barcelona, Spain. [5]Institute for Medical Engineering and Science, and Department of Physics, Massachusetts Institute of Technology, Cambridge, MA, USA. [6]Present address: Institute of Cell Biology, School of Biological Sciences, University of Edinburgh, Roger Land Building, Edinburgh, UK. ✉e-mail: genevieve.almouzni@curie.fr

produced at high levels at S phase entry[23], is deposited in a DNA synthesis-coupled (DSC) manner[24]. This is mediated by the CAF-1 complex[24,25], coupled to replisome progression through its interaction with the DNA polymerase sliding clamp PCNA[26]. In contrast, the replacement variant H3.3 is expressed throughout the cell cycle[27] and incorporated in a DNA synthesis-independent (DSI) manner[24,28]. The first identified chaperone involved in this DSI pathway is the HIRA complex[24,29]. HIRA ensures H3.3 enrichment at active regions, regulatory elements[30,31] and sites of high turnover[32], likely through interacting with RNA Pol II[31]. This H3.3 genomic distribution is paralleled by active PTM patterns[30,33]. Additionally, HIRA deposits H3.3 at transiently exposed DNA in a gap-filling manner[31]. More recently, by mapping de novo incorporation of H3.1 and H3.3 in S phase[19], we revealed that new H3.3 deposition occurred systematically at pre-existing H3.3-enriched sites, while new H3.1 followed the replication fork movement. This dual deposition mechanism results in the establishment of H3.3/H3.1 boundaries. Strikingly, these boundaries overlap with early replication IZs. Notably, HIRA knock-out (KO) disrupted not only these H3.3/H3.1 boundaries, but also the corresponding replication IZs independently of transcription[19], uncovering an unanticipated connection for HIRA and DNA replication. In the absence of HIRA, we distinguished two types of early IZs: (i) those that we called "blurred sites" since both the H3.3 pattern and replication initiation of the early IZs became fuzzy at the boundaries of H3.3 peaks and (ii) those that we called buried sites corresponding to singular H3.3 peaks which disappeared along with abrogation of replication initiation[19]. "Blurred sites" and "buried sites" corresponded respectively to actively transcribed and inactive domains. The next major issue was then to understand which functional relationships could link H3.3 deposition by HIRA and early replication. One possibility is that HIRA and/or H3 variant patterns may contribute to the definition of early IZs by influencing their 3D organisation.

Chromatin has a multi-scale organisation ranging from a basic unit, the nucleosome, up to higher order folding within the nucleus. The nucleosome comprises ~146 bp DNA wrapped around an octamer of four core histones, H2A, H2B, H3 and H4[34] and linker DNA[35], a repeated module that forms the nucleofilament. Beyond the nucleofilament, folding occurs with loops[36] and topologically-associating domains (TADs), each enriched in self-interactions and insulated from neighbouring regions[37–39]. A further level of higher organization is the partitioning into compartments A and B, which are spatially segregated from each other and correspond respectively to open, active euchromatin and dense, repressive heterochromatin[40–42]. Surprisingly, while we have learnt a lot about loop extrusion as mechanism contributing to the dynamics of loops[43,44], the nucleosomal dimension with the assembly pathways, choices of histone variants and their PTMs has not been explored for its impact on genome folding in mammals. While the general histone chaperone FACT was recently shown to impact nucleosome occupancy and active gene organisation in human cells[45], we still do not know whether and how HIRA-mediated H3.3 incorporation at active chromatin impacts its organisation in 3D. This is particularly intriguing considering early mammalian development, when higher-order chromatin organisation is being established[46–48] concomitantly with a genome-wide H3.3 redistribution[49]. Therefore, we hypothesized that HIRA-mediated nucleosome assembly could also have an important function in the organisation of active chromatin, including early replication IZs.

Here, we used knock-out (KO) and rescue experiments of the histone H3.3 chaperone HIRA to dissect the relationship between early replication initiation, and the combination of chromatin state including histone variant H3.3, histone PTMs, and 3D genome organisation. First, we demonstrated that HIRA ensures enrichment of H3.3 within compartment A and limits its accessibility. In the absence of HIRA, we detect weaker A-A compartment interactions and minor compartment A to B (A-to-B) switching without a redistribution of H3 PTMs on this scale. Focusing on early IZs, we found that HIRA plays a role in their definition independently from its impact on accessibility and without affecting their H3 PTM pattern. Furthermore, only a fraction of HIRA-dependent non/low-transcribed early IZ (buried sites) belong to compartment A in wild-type (WT) cells, and in the absence of HIRA half of them switched to B as they lost capacity for early initiation. Yet, rescue with HIRA recovered the H3.3 enrichment and early firing patterns at early IZs without necessarily restoring their compartment A identity. Thus, our results indicate that HIRA is important for H3.3 enrichment, accessibility and higher-order organisation of active chromatin, while it defines early IZs independently of their accessibility, PTM pattern and compartment organisation.

## Results

### HIRA ensures proper provision of H3.3 in compartment A and restricts its accessibility

Considering the importance of the H3.3-specific chaperone HIRA for early replication[19], we wondered whether it could relate to an impact of histone variant deposition on higher-order chromatin organisation. We thus used our previously characterized constitutive knock-out (KO) of HIRA in HeLa cells[50] and compared 3D genome organization in wild-type (WT) or HIRA KO cells by Hi-C. We assessed in parallel the distribution of H3.3, H3.1, chromatin accessibility and a set of H3 PTMs (Fig. 1a). To avoid S phase heterogeneity due to their distinct incorporation[51], we used the data from G1/S cells[19] to profile H3 variant enrichment. First, we confirmed that Hi-C maps from the two parental H3.1-SNAP and H3.3-SNAP cell lines showed comparable compartment calls, based on eigenvector (EV) decomposition[40] at 50 kb resolution and cis/trans contact ratios (Supplementary Fig. 1a, 1b). We found that compartment A showed a strong enrichment in H3.3 with depletion of H3.1 (Fig. 1b). This was in contrast with compartment B which was relatively depleted in H3.3 and enriched in H3.1 only in large (>2 Mb) domains (Fig. 1b). We then compared Hi-C maps obtained from WT cells with HIRA KO (Fig. 1b, c, left). The two conditions had similar cis/trans contact ratio (Supplementary Fig. 1a) and contact distance decay (Supplementary Fig. 1c) but also displayed some distinct features. Indeed, the proportion of compartment changes from WT to HIRA KO (2.1% A-to-B and 1% B-to-A, Fig. 1d) while limited was significant since it exceeded variability between the cell lines (Supplementary Fig. 1b). Furthermore, saddle-plot analysis revealed reduced contact frequency between A-A and A-B in parallel with increased interactions between B-B regions in cis in HIRA KO (Supplementary Fig. 1f). Finally, identification of TADs at 10 kb resolution revealed that TAD borders overlapped between WT and HIRA KO (Supplementary Table 1) to the same extent as between H3.1- and H3.3-SNAP cell lines. Thus, HIRA KO contributes to limited but significant changes in compartment identity and affects interactions in compartment A without impairing TAD-scale organisation.

In the absence of HIRA, H3.3 enrichment decreased throughout compartment A, whereas H3.1 did not show substantial changes (Fig. 1b, c, quantified in Supplementary Fig. 1d, e). At compartment borders, the transition for both variants became fuzzy, reminiscent of our previously reported blurring at H3.3-enriched sites[19]. Given the striking redistribution of H3.3 in HIRA KO and the previously reported increased DNase I sensitivity following transient HIRA depletion[31], we compared chromatin accessibility in WT and HIRA KO cells arrested in G1/S using ATAC-seq (Fig. 1a). While ATAC-seq peaks overlapped in the two conditions (Supplementary Table 2), we detected a consistent increase in the number of peaks when comparing WT to HIRA KO. On the scale of compartments, in the absence of HIRA, accessibility increased in A but diminished in B (Fig. 1e), contrasting the redistribution of H3.3. Thus, we conclude that lack of HIRA leads to changes in higher order organization showing defects at a compartment level. This comprises a general decrease of H3.3 enrichment in A compartment, a concomitant increase in accessibility and decrease in contacts in compartment A but a minor effect on compartment identity.

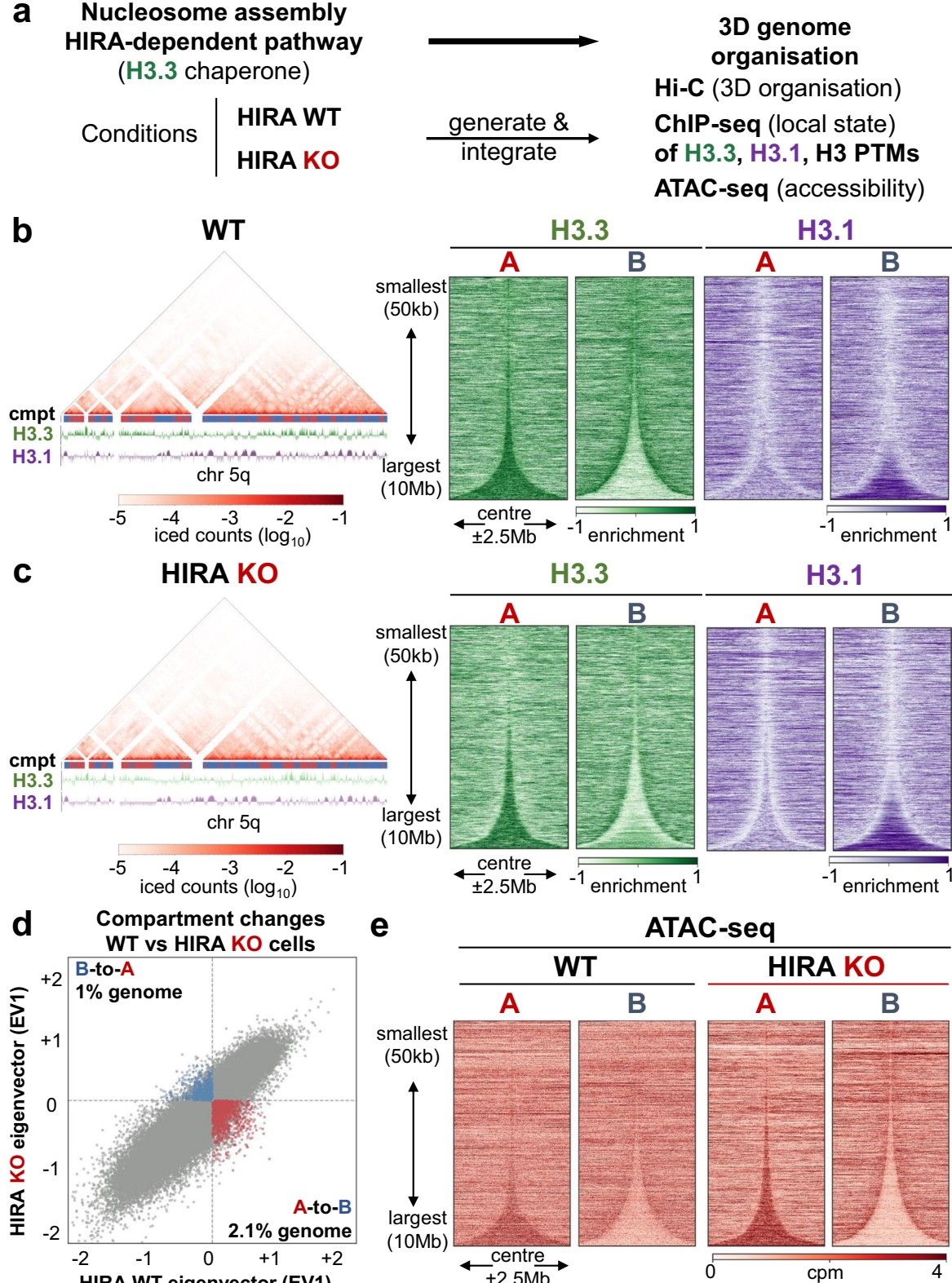

**At the compartment scale, H3.3 redistribution in the absence of HIRA is not accompanied by corresponding PTM changes**

Given the reports showing that H3.3 in chromatin has been associated with the presence of active marks[30,33], and that phosphorylation of the H3.3-specific S31 residue can promote H3K27ac deposition by p300[52–54] or inhibit H3K9me3 removal by KDM4B[55], we investigated whether the redistribution of H3.3 in the absence of HIRA led to changes in H3 PTMs. We performed native ChIP-seq for a panel of selected active (H3K4me3, promoter-associated, H3K4me1, H3K27ac, enhancer-associated) and inactive (H3K9me3, constitutive heterochromatin and H3K27me3, facultative heterochromatin) H3 marks (Fig. 1a). When comparing to H3.3, patterns obtained for each of these marks in HIRA WT or KO did not reveal changes of PTM patterns in A/B compartments that mirror the H3.3 redistribution (Fig. 2). Thus, a

**Fig. 1 | HIRA ensures proper provision of H3.3 and restricts accessibility in compartment A. a** Experimental strategy to assess the effect of disrupting chromatin assembly on higher-order genome organization by constitutive HIRA knockout (KO). Hi-C was performed in WT and HIRA KO H3.1-SNAP and H3.3-SNAP cells. To compare the 3D folding of chromatin to local state, we obtained H3.1 and H3.3 SNAP ChIP-seq from Gatto et al. (2022)[19] and profiled H3 PTMs by ChIP-seq and accessibility by ATAC-seq. **b** Left: Representative Hi-C map at 50 kb resolution with A/B compartment track and H3.1 and H3.3 enrichment at 10 kb resolution of chromosome 5q (chr5: 50–170 Mb) from WT cells. Right: Total H3.3 and H3.1 enrichment at 10 kb bins at A ($n = 1573$) and B ($n = 1573$) compartment domains from WT HeLa cells, sorted by size and centered at their middle ± 2.5 Mb. **c** Left: Hi-C map, compartment track and right: H3.1 and H3.3 enrichment shown in A ($n = 1582$) and B ($n = 1588$) compartment domains as in (**b**), for HIRA KO cells. **d** EV1 (1st eigenvector, indicating compartment) of 50kb-binned Hi-C matrices from HIRA WT vs KO cells. Bins which change from A-to-B (lower right quadrant) or B-to-A (upper left quadrant) in the same direction in both cell lines are coloured red and blue, respectively. **e** ATAC-seq at 10 kb bins at compartments A and B from WT and HIRA KO HeLa cells, sorted by size and centered at their middle ± 2.5 Mb ATAC-seq is shown as cpm. H3.3 and H3.1 enrichment shown is z-score of $\log_2$ IP/input.

disconnection between the variants and the associated marks could be revealed. More specifically, where we observed an increase in H3.3 enrichment (Fig. 1c, Supplementary Fig. 1d), no gain of active or loss of inactive marks in large B compartments occurred (Fig. 2a). Thus, we conclude that HIRA while changing H3.3 distribution contributes to compartment organisation independently of the relative H3 PTM enrichment.

### HIRA defines early replication IZs independently of their accessibility and H3 PTM enrichment

We then went on and examined how changes in chromatin accessibility and higher-order organisation in the absence of HIRA could relate to change in early replication initiation. In the HIRA KO model, we previously characterised defects in entry and progression into early S phase by monitoring EdU incorporation 2 h after G1/S release[19]. We further confirmed here that we could reproduce these defects after transient depletion of HIRA in HeLa and U2OS cells (Supplementary Fig. 2). In this way we first ensure that the replication defect in HIRA KO cells cannot reflect a mere adaptation to a constitutive loss of the chaperone and further show that it also occurs in another cell type with defect in ATRX (another H3.3 chaperone[30,56]). Given the links between early replication, H3.3[53], H3 marks[12] and open chromatin, we first wished to examine if HIRA affected early IZ firing by modulating their local chromatin state. Since we previously identified and coined blurred and buried sites for early replication initiation zones (IZs) based on the defects observed after HIRA KO[19] (Fig. 3a for schematic illustration), we considered these two types of sites. In WT cells, both blurred and buried sites showed high accessibility at their boundaries (corresponding to early IZs) (Fig. 3b, Supplementary Fig. 3a), enrichment in H3K4me1 and H3K27ac and depletion in H3K9me3 and H3K27me3 (Fig. 3c). At the boundaries of blurred sites, we found an enrichment in H3K4me3 which was not present at buried sites, in line with the presence of active transcription at blurred sites[19]. In the absence of HIRA, similar to the increase observed in compartment A (Fig. 1e), ATAC-seq signal increased within blurred sites (Fig. 3b, right) but they maintained the sharpness at their boundaries (Supplementary Fig. 3a). This latter behaviour does not mirror the blurring of H3.3 enrichment, EdU 2 h signal (Fig. 3b) and H3.3/H3.1 ratio (Supplementary Fig. 3b). At buried sites, accessibility barely decreased in the absence of HIRA and maintained its pattern (Fig. 3b, Supplementary Fig. 3a). These data were in sharp contrast with the complete loss of H3.3 enrichment and early firing (Fig. 3b) and did not match changes in expression (Supplementary Fig. 3b). Furthermore, we did not detect reduced precision or drastic redistribution of the marks matching the changes in H3.3 and early firing at either type of sites (Fig. 3c, Supplementary Fig. 4a). These data are in line with our results for A/B compartments where we observed a disconnection between H3.3 redistribution, marks and ATAC-seq. To further confirm this observation, we also analysed H3.3-rich early IZs identified by OK-seq at 1 kb resolution[7,19]. Indeed, both H3.3 and EdU 2 h enrichment blurred and decreased at non/low-expressed H3.3+ OK-seq IZs without a corresponding change in ATAC-seq in the absence of HIRA (Supplementary Fig. 3c). In contrast, accessibility increased at early IZs flanked by transcribed sites scaling with their expression level, while H3 PTMs remained unchanged across

all sites (Supplementary Fig. 4b). Thus, we can conclude that HIRA regulates early firing at both blurred and buried sites independently of chromatin accessibility and histone PTMs.

### In the absence of HIRA, only non-transcribed early IZs switch from compartment A to B

In addition to chromatin state, early firing also correlates with features of higher-order genome organisation with early replicating regions generally corresponding to compartment A[10]. Given the impact of HIRA for compartment A organisation (Fig. 1), we wondered whether it can regulate firing at early IZs by influencing their compartment identity. We found that blurred sites generally were located in compartment A in both WT (85% sites) and HIRA KO cells (86% sites, Fig. 4a, green points). This was also reflected by their positive EV1 values in both conditions (Fig. 4b) and absence of substantial compartment switching compared to random sites (Fig. 4c). In contrast, buried sites were located mainly in compartment B in WT (60% sites, Fig. 4a, red points) with a small fraction in compartment A (40% sites). Furthermore, buried sites significantly switched compartment in HIRA KO (23.5% sites, Fig. 4c, Supplementary Fig. 5a), predominantly from A to B (20% sites, Fig. 4d). Given the association of compartment A with gene expression, we examined transcription at buried sites by RNA-seq to determine if it could explain their behaviour. In WT cells, buried sites could be found in either compartment A or B regardless of their expression (Fig. 4d). Buried sites which switched from A to B had significantly lower expression in WT cells than those that remain in compartment A in the absence of HIRA (Fig. 4d, Supplementary Fig. 5b). However, A-to-B switching in HIRA KO was not accompanied by a significant decrease in transcription (Supplementary Fig. 5b, c). Thus, our data reveal that early IZs, which are transcribed (blurred sites) are located in compartment A irrespectively of the HIRA status. In contrast, low/non-transcribed early IZs (buried sites) are found predominantly in compartment B with a small fraction in compartment A in WT cells. These findings indicate first that early initiation does not always occur in compartment A in differentiated cells, and second, that only non-transcribed buried sites showed a significant switch from compartment A to B along with loss of early firing in the absence of HIRA.

### HIRA rescue reestablishes H3.3 pattern and early replication initiation at both blurred and buried sites

To disentangle links between H3.3 deposition, early replication and compartment assignment, we decided to test if we could restore both timely firing at early replication sites and compartment organization, we performed HIRA rescue experiments. We transiently transfected HIRA-YFP (HIRA) or YFP only (control) plasmid for 48 h in H3.1-SNAP and H3.3-SNAP HIRA KO cells followed by G1/S-synchronization (Fig. 5a), obtaining high efficiency (>70%) in all conditions (Supplementary Fig. 6a). We then evaluated H3.3 and H3.1 distribution at blurred and buried sites by SNAP-Capture ChIP-seq. Rescue with HIRA, but not with control plasmid, restored both the pattern and levels of H3.3 enrichment at blurred sites (Fig. 5b, c, top, Supplementary Fig. 6b). Strikingly, HIRA complementation was also sufficient to target H3.3 incorporation to buried sites, despite their complete loss of H3.3

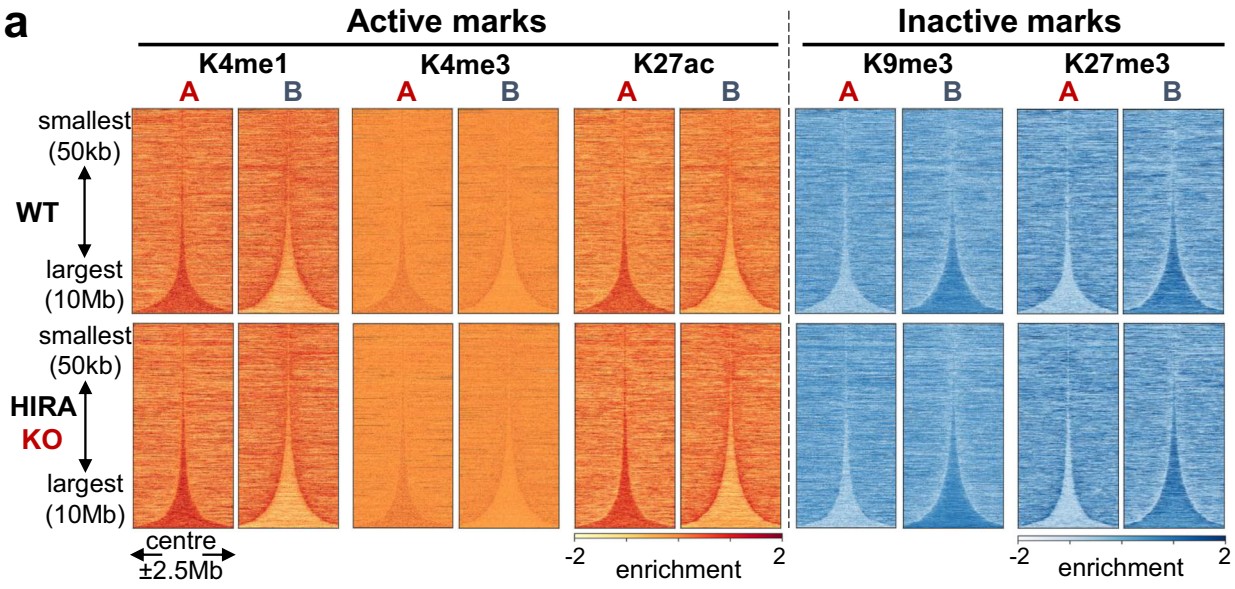

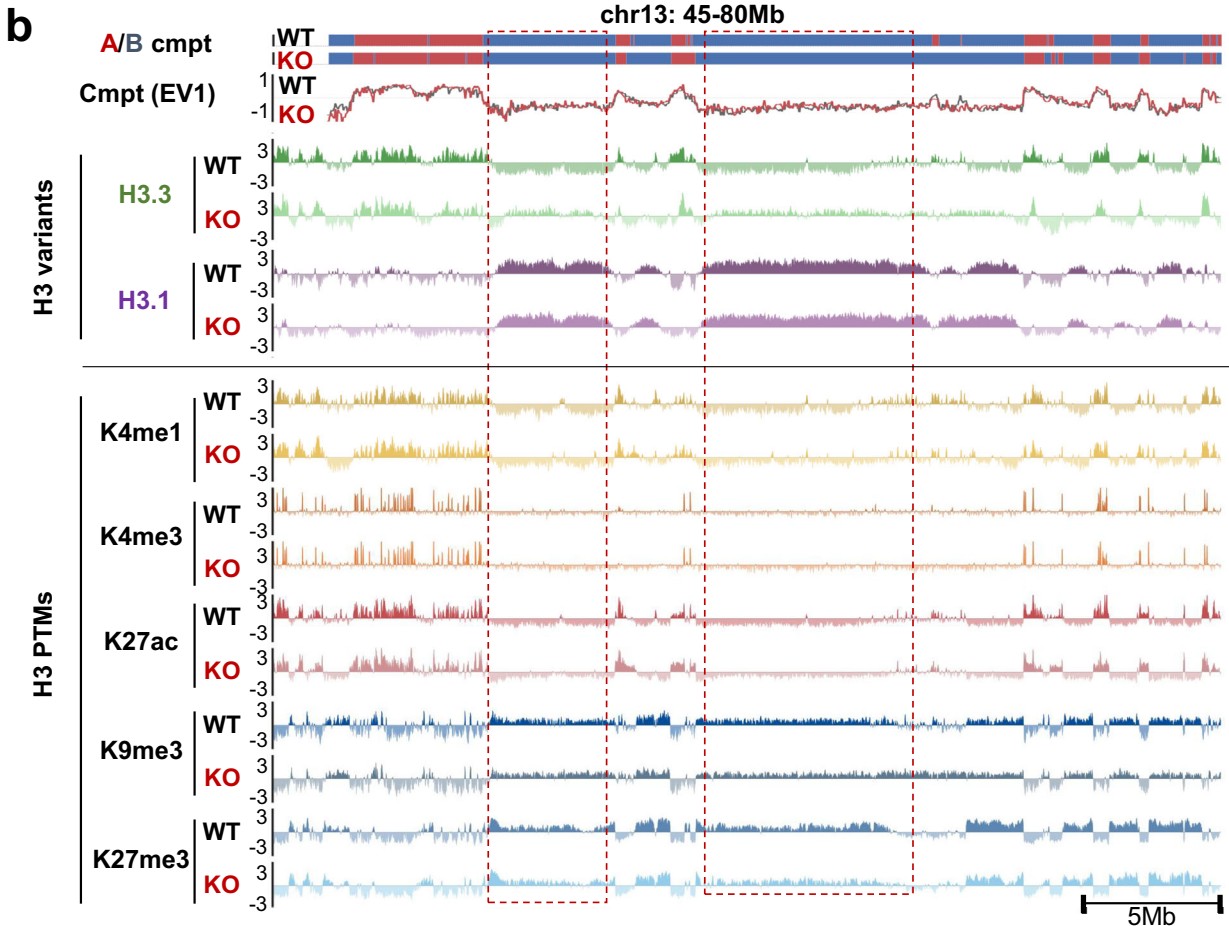

**Fig. 2 | At the compartment scale, H3.3 redistribution in the absence of HIRA is not accompanied by corresponding H3 PTM changes. a** Active (H3K4me1/3, H3K27ac) and inactive (H3K9/27me3) PTM enrichment at 10 kb bins at compartments A and B from WT and HIRA KO HeLa cells, sorted by size and centered at their middle ±2.5 Mb. **b** Compartment assignment and eigenvector tracks at 50 kb resolution and enrichment of H3.3, H3.1, active (H3K4me1/3, H3K27ac) and repressive (H3K9/27me3) PTMs from HIRA WT and KO cells (chr13: 45–80 Mb). ChIP-seq is shown at 10 kb bins smoothed over 3 non-zero bins. H3.3, H3.1, PTM enrichment shown is z-score of $\log_2$ IP/input.

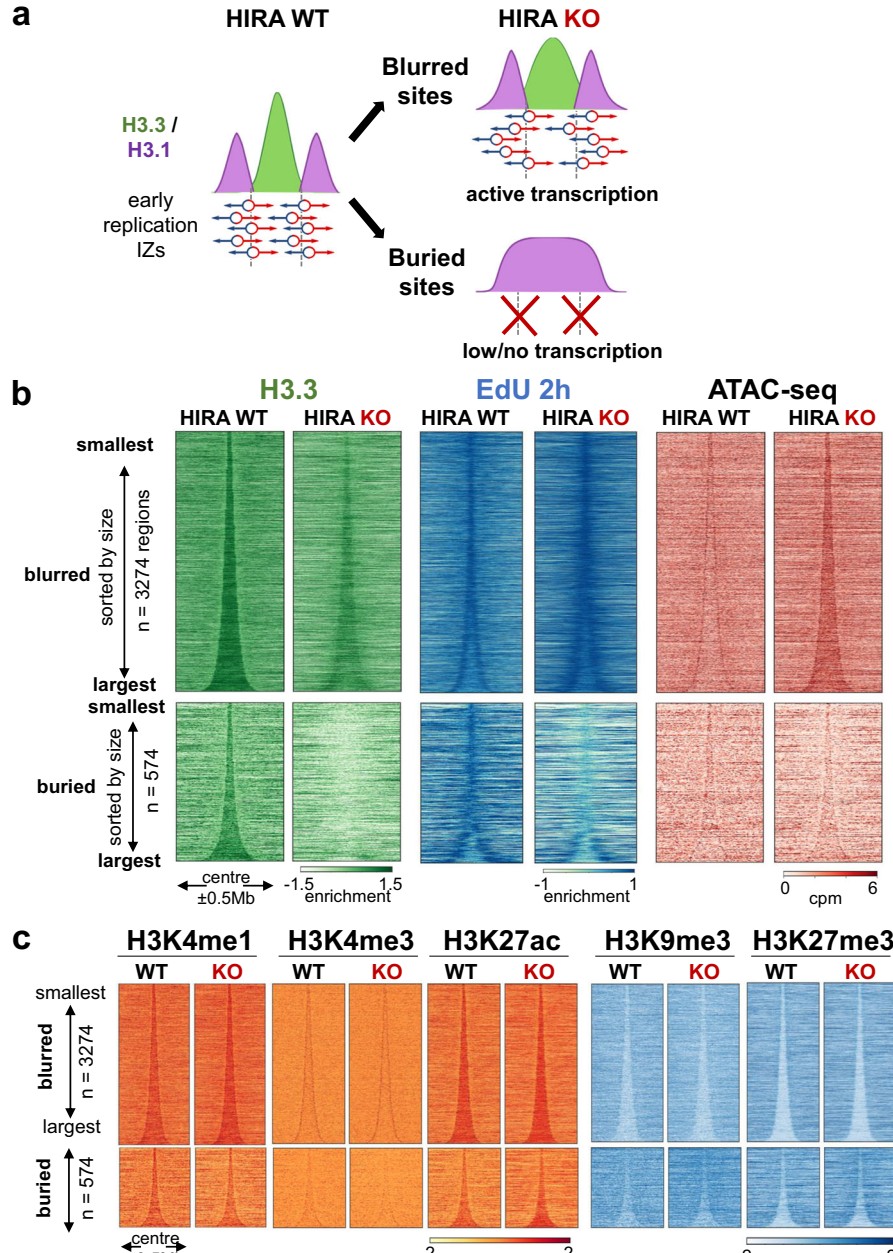

**Fig. 3 | HIRA defines early replication initiation zones independently of their accessibility and H3 PTM enrichment. a** Schematic representation of early replication initiation zones defined by HIRA-dependent H3.3/H3.1 boundaries (adapted from Gatto et al., 2022[19] with permission from Elsevier). **b** Enrichment of H3.3, EdU at 2 h in S and ATAC-seq signal from WT and HIRA KO cells at blurred (*n* = 3274, top) and buried sites (*n* = 574, bottom), sorted by size and centered at their middle ± 0.5 Mb. **c** Active (H3K4me1, H3K4me3, H3K27ac) and repressive (H3K9me3, H3K27me3) histone PTM enrichment profiles from WT and HIRA KO cells at blurred (*n* = 3274, top) and buried (*n* = 574, bottom) sites, centered at their middle ± 0.5 Mb and sorted by size. Enrichment of H3.3, H3 PTMs and EdU is shown as z-score of log₂ IP/input, ATAC-seq is shown as cpm at 10 kb bins.

upon HIRA KO and low or absent transcriptional activity (Fig. 5b, c, bottom, Supplementary Fig. 6b). Concomitantly, we also detected recovery of the H3.1 enrichment and the H3.3/H3.1 ratio (Supplementary Fig. 6c, d). Thus, re-supplying HIRA is sufficient to reestablish H3.3 enrichment and H3.3/H3.1 balance at blurred and buried sites.

Next, we examined whether the recovery of the H3 variant pattern could restore firing at pre-existing replication IZs in early S phase. To test this, we monitored early replication in synchronized cells by following DNA synthesis using either EdU incorporation[19] or PCNA staining. Upon rescue of the HIRA KO cell line with HIRA, the proportion of cells in S phase 2 h after G1/S release significantly increased,

suggesting a recovery of early initiation (Supplementary Fig. 6e). EdU-seq at 2 h after release in S revealed early firing again restricted within blurred sites and significantly increased at buried sites (Fig. 5d, e, Supplementary Fig. 6f, g). Thus, our data are consistent with a restoration of early replication firing patterns concomitant with the re-establishment of H3.3/H3.1 balance.

**HIRA rescue recovers H3.3 pattern and early initiation at buried sites without compartment reversal**

Finally, we performed Hi-C to assess if we also recovered genome organization upon HIRA rescue. We transfected HIRA KO cells with

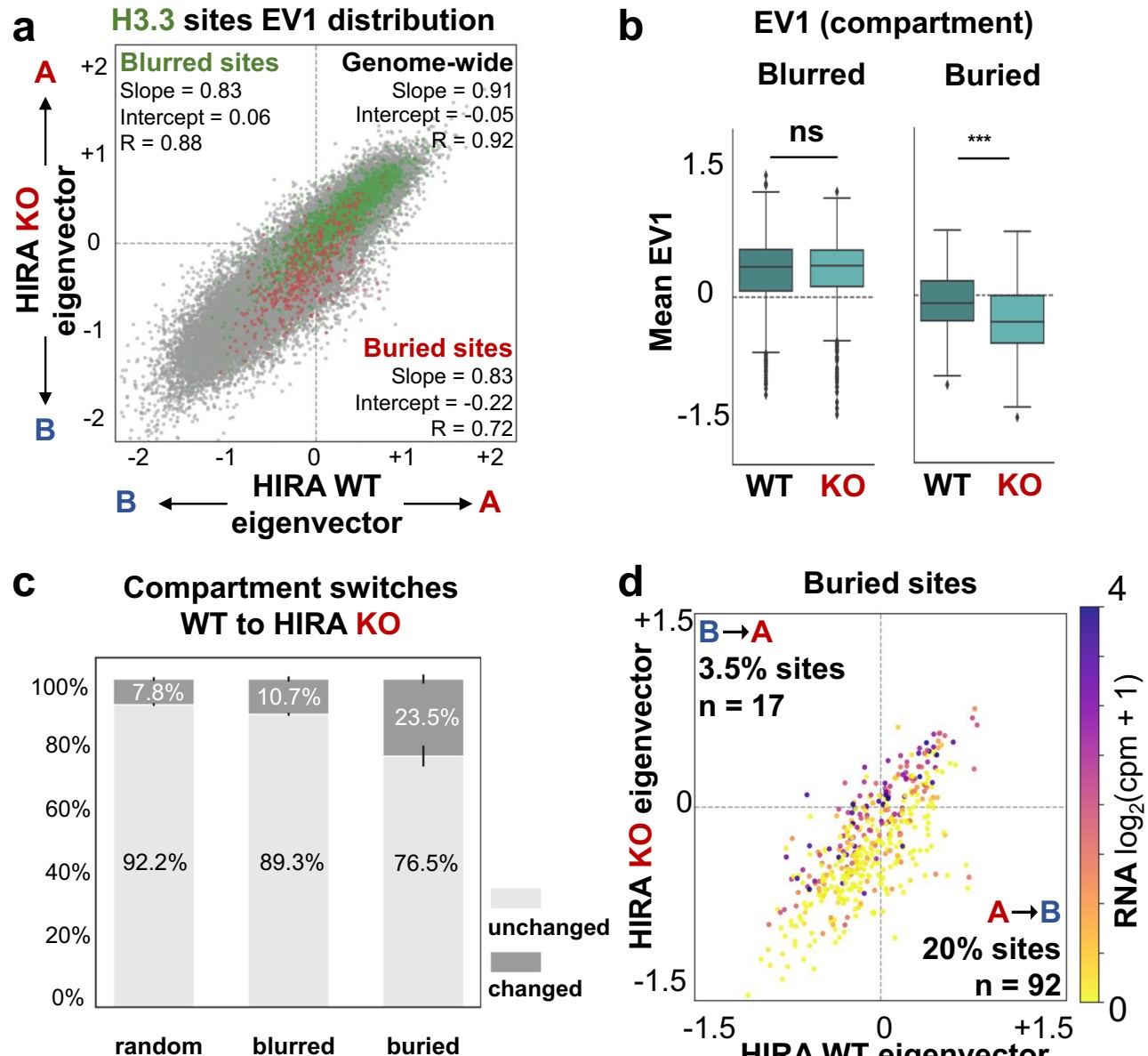

**Fig. 4 | In the absence of HIRA, only non-transcribed early IZs switch from compartment A to B. a** Comparison of EV1 distribution genome-wide (50 kb bins, $n = 40887$, grey) or at blurred ($n = 2382$, green) and buried ($n = 439$, red) sites between WT and HIRA KO cells. Slope, intercept and R from linear regression are noted for each set. **b** Mean value of EV1 (1st eigenvector, indicating compartment) at blurred ($n = 2382$) and buried ($n = 439$) sites from WT (teal) and HIRA KO (light teal) cells. The boxplot centre is the median, the bounds of the box are the first and third quartiles and the whiskers extend to 1.5x IQR. Source data are provided as a Source data file. **c** Proportions of blurred and buried sites which remain in the same compartment (unchanged, light grey) or undergo a switch (changed, dark grey) from WT to HIRA KO cells. A set of randomised size-matched sites was quantified as

control. **d** Scatterplot of mean EV1 value at buried sites ($n = 439$) from WT and HIRA KO cells. Colour represents transcriptional activity from HIRA WT cells. Proportion of sites which change from compartment A-to-B (lower right quadrant) or B-to-A (upper left quadrant) are quantified. EV1 was calculated from 50 kb-binned Hi-C matrices and then re-binned to 10 kb to compute mean EV1 value for blurred and buried sites. Transcriptional activity is measured as the mean $\log_2$(cpm+1) RNA-seq signal binned at 10 kb for each site. Two-tailed Mann-Whitney U test corrected for multiple testing by FDR (5% cut-off) was used to determine significance of differences between WT and HIRA KO. Significance was noted as: * ($p < =0.05$), ** ($p < =0.01$), *** ($p < =0.001$) for all comparisons. *P*-values HIRA WT vs KO at blurred sites $p = 0.45$, at buried sites $p = 5.14e^{-14}$.

HIRA-YFP (HIRA) or YFP (control) plasmid (Supplementary Fig. 7a) and validated the re-establishment of H3.3 enrichment in compartment A (Supplementary Fig. 7b) at blurred and buried sites by SNAP-capture ChIP-seq (Supplementary Fig. 7c). Hi-C maps from control (YFP) and HIRA rescue recovered A compartment interactions (Supplementary Fig. 7d) but showed very few compartment switches (Fig. 6a, grey dots), despite the restoration of H3.3 balance in compartments. This was echoed by the behaviour of both blurred and buried sites (Fig. 6a, green and red dots, respectively). While buried sites switched more

than expected by chance (7.7%, Fig. 6b), only 5.3% switched back from compartment B to A upon HIRA rescue (in contrast to 20% A to B upon HIRA KO). Furthermore, HIRA rescue recovered both H3.3 enrichment and early firing in buried sites remaining in B regardless of whether they had previously switched compartment from WT to HIRA KO (Fig. 6c). Finally, following HIRA rescue, the restoration of H3.3 enrichment and early initiation in buried sites occurred to similar extents whether they switched from B to A or remained in B (Fig. 6d, Supplementary Fig. 7e). We thus conclude that recovery of early

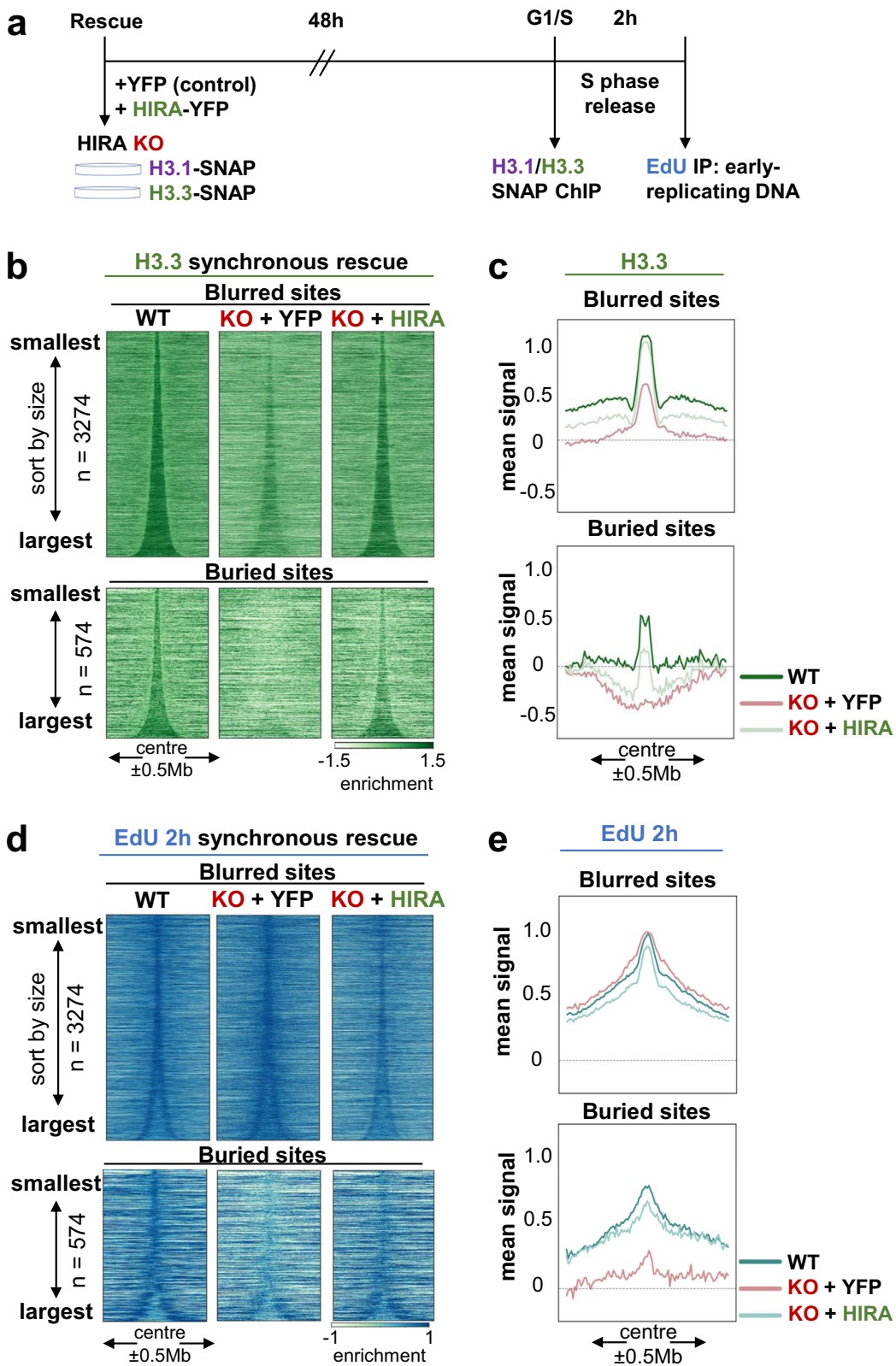

initiation zones at buried sites occurs irrespective of initial compartment, and when a switch had occurred there was no need to switch back to be able to detect the early IZ.

## Discussion

In this study, to understand how the chaperone HIRA responsible for H3.3 deposition could affect early replication initiation, we simultaneously explored its impact on local chromatin state, histone marks, and higher-order genome folding (Fig. 7). First, by combining data from Hi-C, ChIP-seq and ATAC-seq experiments, we revealed that the absence of HIRA leads to a loss of H3.3 enrichment in compartment A. This is accompanied by changes in its accessibility and 3D organisation, without an impact on histone mark enrichment (Fig. 7a). Second, we examined the functional importance of HIRA in defining early

**Fig. 5 | HIRA rescue reestablishes H3.3 pattern and early replication initiation at both blurred and buried sites. a** Scheme of experimental strategy to perform HIRA rescue combined with G1/S synchronization to assay total H3.1/H3.3-SNAP by ChIP-seq and new DNA synthesis in early S phase (2 h release). Asynchronous cells constitutively expressing H3.1- or H3.3-SNAP were transfected with YFP (control) or HIRA-YFP plasmid. Cells were then arrested at the G1/S boundary by double thymidine block (starting 6 h post-transfection). Total H3.1- and H3.3-SNAP were assayed by SNAP-Capture ChIP-seq of native MNase-digested chromatin, with matching inputs collected. For EdU-seq, cells were released in S phase for 1.5 h, followed by 30 min EdU pulse and collection at 2 h in S phase, followed by EdU IP. **b** H3.3 and (**d**). EdU at 2 h in S enrichment profiles from WT (as reference) and HIRA KO rescue with YFP (control) and HIRA plasmid at blurred ($n = 3274$) and buried sites ($n = 574$), sorted by size and centered at their middle ±0.5 Mb. **c** H3.3 and (**e**). EdU at 2 h in S mean signal at blurred and buried sites between 60-160 kb in length, centered in their middle ± 0.5 Mb, corresponding to the conditions described above. Enrichment relative to input was calculated at 10 kb bins as z-score of $\log_2$ IP/input.

replication initiation zones (Fig. 7b). We found that changes in early initiation did not strictly follow changes in chromatin accessibility and histone PTMs in HIRA KO. Next, we combined deletion and rescue experiments and revealed that HIRA defines early IZs irrespectively of their compartment organisation. We discuss implications for (i) disentangling the relationship between H3 variants, chromatin state and A compartment organisation and (ii) our understanding of how early replication IZs are defined and the mechanism of HIRA recruitment to these sites.

## HIRA-mediated H3.3 nucleosome assembly promotes compartment A contacts independently of PTMs

Compartment A showed a decrease in H3.3 enrichment in HIRA KO cells, along with a decrease in A-A interactions and a modest but significant (2.1%) proportion of A-to-B switches (Fig. 7A). Several studies have provided support to the idea that histone PTM states may contribute to compartmentalisation[57,58], although the mechanism remains unclear[59]. The lack of H3 PTM redistribution at this scale in our system was somewhat unexpected given reports of the role of phosphorylation of the unique H3.3S31 residue in promoting H3K27ac at regulatory elements[53,54]. This data highlights that although H3.3 and enhancer-associated PTMs may be functionally linked at regulatory regions, changes in their distribution at this genomic scale does not necessarily translate into reorganisation with respect to larger domains. Previous work reported that modifications on both H3.1 and H3.3 from oligonucleosomes show comparable features and rather relate to the chromatin environment[60]. This suggests that at the scale of compartments, the distribution of H3 variants and PTMs can be disconnected. Furthermore, the observed decrease of interactions in compartment A implies that they are influenced not only by the histone marks, as widely accepted[59]. We propose here an important role for nucleosome organisation—the choice of histone variants—and density, especially given the increase in ATAC-seq observed, for compartment A organisation. This is in agreement with recent work showing that interactions between genes and regulatory (broadly H3K27ac) regions contribute to compartmentalisation[61–63], in addition to the major role of high affinity heterochromatin interactions[57]. Furthermore, it hints at the possibility that active marks may be functionally related to genome organisation not just via their readers but also through changes in nucleosome dynamics and/or accessibility. Thus, our data helps disentangle the long-standing correlation between H3.3, histone PTMs and A/B compartments, disconnecting PTMs from histone variant accumulation. We anticipate that this view can stimulate new avenues to explore histone variant dynamics at distinct sites or developmental stages in normal or disease states and the links with higher-order chromatin organisation.

## HIRA defines early replication IZs independently of chromatin accessibility and histone marks

In line with other reports examining early initiation[9], we found in WT cells that HIRA-dependent early IZs at the flanks of both blurred and buried sites show high accessibility. In the absence of HIRA, accessibility increases within blurred and weakly decreases in buried sites. While we do detect changes, they do not match the loss of precision or detection of early initiation, respectively. Thus, we propose that the

role of HIRA in defining early IZs is independent of its role in regulating chromatin accessibility. Furthermore, our results also underline the fact that IZ accessibility on its own is not sufficient to ensure precision (at blurred sites) or timing (at buried sites) of early initiation. In addition to accessibility, a defined set of histone marks have also been associated with early IZs[64]. Here, we found that both blurred and buried sites are enriched in active (H3K4me1/3, H3K27ac) and depleted in inactive (H3K9me3, H3K27me3) PTMs. Strikingly, these patterns remained essentially unchanged in the absence of HIRA, which parallels our findings on the scale of compartments, unlike H3.3 and early initiation. Indeed, the prediction of early replication based on active marks or accessibility, although highly correlated, is less robust than the use of H3.3[65,66]. Here, we have demonstrated that the early initiation defect upon HIRA loss mirrored the impaired H3.3 distribution rather than accessibility or PTM patterns. In conclusion, our data put forward a key role for H3.3 deposition by HIRA in defining early IZ firing independently of the histone marks we assayed.

## HIRA rescue enables recovery of H3 variant enrichment defining sharp boundaries and early firing at IZs irrespective of compartment

The majority of HIRA-dependent early IZs are located at the boundaries of what we had previously coined blurred sites. We find these mainly in compartment A, where they remain in the absence of HIRA despite the reduced precision in both H3.3 enrichment and early replication[19]. At blurred sites, HIRA rescue reestablishes a sharp boundary of H3.3/H3.1 and restores early initiation profiles, again without changes in compartment. It is thus tempting to speculate that HIRA recruitment in blurred sites could be mediated by its interaction with RNA Pol II or via its 'gap-filling' mechanism[31], due to the presence of active genes and increased accessibility. In contrast, early initiation also occurs at the boundaries of buried sites which absolutely require HIRA for early initiation and in WT cells, are predominantly in compartment B. Strikingly, HIRA rescue recovered both H3.3 enrichment and early initiation at buried sites despite their low/no transcriptional activity. Here, it is important to envisage how HIRA is recruited back given that we cannot invoke a transcription-linked marking as above or compartment organisation (discussed in more detail below). The fact that we could recover H3.3 enrichment within buried sites implies that (i) these regions remain 'bookmarked' in the absence of HIRA and (ii) this is sufficient to guide HIRA back following rescue. On the one hand, given the capacity of HIRA to directly bind DNA[31,67], an attractive possibility is that buried sites may have particular DNA properties which mediate this recruitment. Indeed, although replication initiation in metazoans does not occur at DNA defined sites, features like G-rich elements can promote early firing[64]. On the other hand, the 'memory' of buried sites may relate to the maintenance of PTMs, another variant (e.g. H2A.Z[21]) or accessibility in the absence of HIRA. Additionally, HIRA is a chaperone complex comprising three subunits: HIRA[68], UBN1/2[24,69], CABIN1[24,70,71]. Although their stability requires the HIRA protein[31], UBN1 and CABIN1 remain detectable in the nuclei of HIRA KO cells[50]. Since UBN1 can bind DNA[72], even small amounts retained locally could provide an alternative for 'memory' of the sites. Finally, we cannot exclude the possibility that HIRA may have a role independent of its capacity for H3.3 deposition, as has been demonstrated in the context of

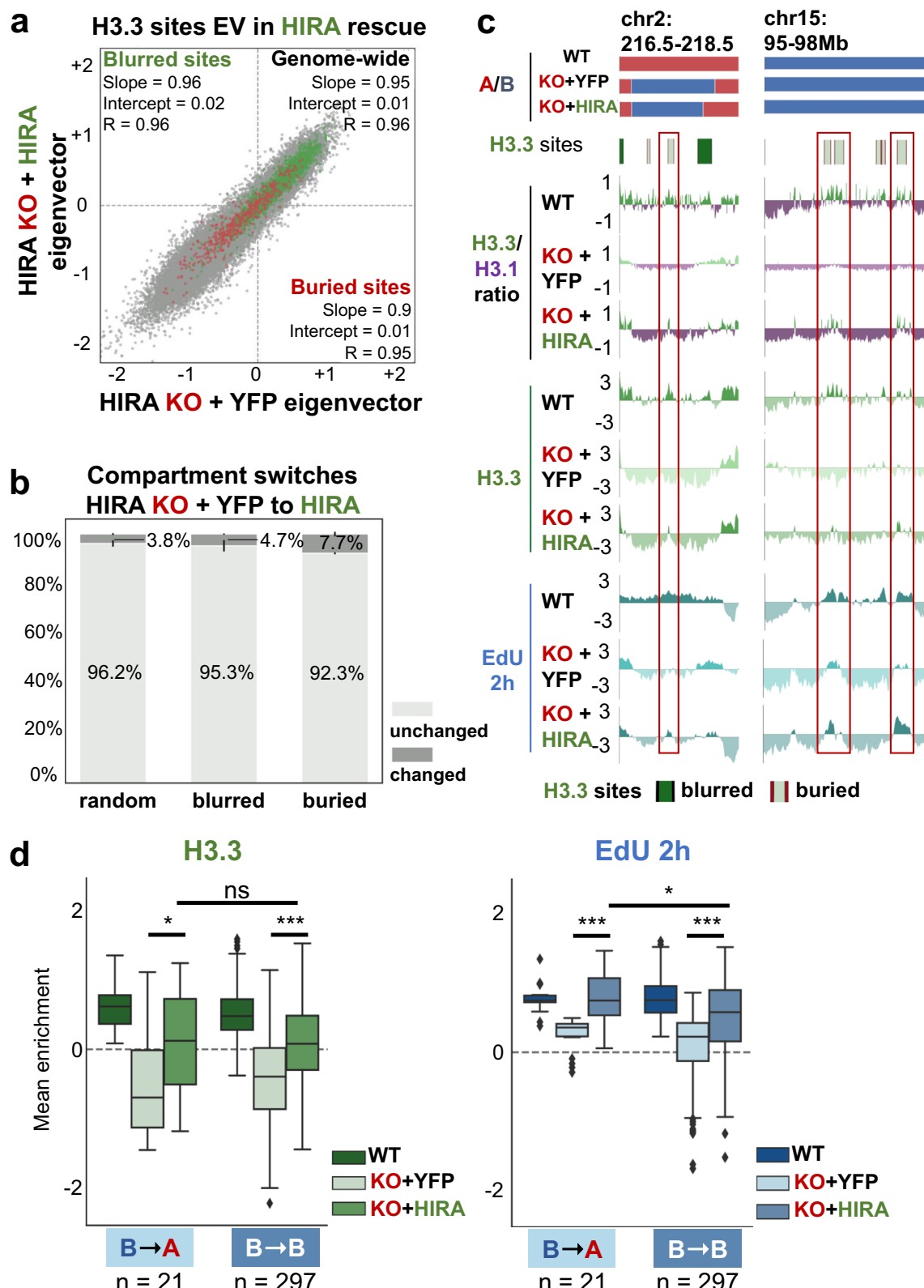

transcriptional restart post-DNA repair[73]. Overall, our data demonstrates that HIRA-mediated targeting of H3.3 is successfully restored upon rescue at both types of pre-existing H3.3 sites. While recovery at blurred sites may rely on RNA Pol II, recovery at buried sites occurs independently of transcription. Importantly, we should emphasize that regardless of the type of IZ considered, H3.3 and early initiation recovery is independent of compartment organisation.

**HIRA-mediated H3.3 deposition at early IZs as a model to dissect the relationship between replication initiation control and 3D genome organisation**

Disentangling the link between early/late replication and A/B compartments has proven challenging due to our limited understanding of the factors that govern them[10,59]. Here, we show that in the absence of HIRA, half of the buried sites previously in compartment A switch to B.

**Fig. 6 | HIRA rescue recovers H3.3 pattern and early initiation at buried sites without compartment reversal. a** Comparison of EV1 distribution genome-wide (50 kb bins, $n = 40887$, grey) and at blurred ($n = 2382$, green) and buried ($n = 439$, red) sites between HIRA KO + YFP (control) and HIRA rescue. Slope, intercept and R from linear regression are noted for each set. **b** Proportions of blurred and buried sites which remain in the same compartment (unchanged, light grey) or undergo a switch (changed, dark grey) from HIRA KO + YFP (control) to HIRA rescue. A set of randomised size-matched sites was quantified as control. **c** Compartment assignment, H3.3 site location (blurred/buried in dark/light green, respectively), H3.3/H3.1 ratio and enrichment of H3.3 and EdU 2 h at representative regions from WT (as reference), HIRA KO rescue with YFP (control) and HIRA. Shown are a set of buried sites which shift from A-to-B after HIRA KO and remain B after rescue (left, cf. Supplementary Figs. 3c, 5c) and a set of buried sites which are always in compartment B (right, cf. Figs. 1b, 2a). Note that EdU 2 h rescue is detected specifically at buried sites which increase H3.3 enrichment and H3.3/H3.1 ratio. **d** Mean H3.3 and EdU 2 h in S enrichment from WT (as reference) and HIRA KO rescue with YFP

(control) and HIRA cells at buried sites that switched from B-to-A ($n = 21$) or remained in B ($n = 297$) from HIRA KO rescue with YFP (control) to HIRA. The boxplot centre is the median, the bounds of the box are the first and third quartiles and the whiskers extend to 1.5x IQR. Source data are provided as a Source data file. Enrichment relative to input was calculated at 10 kb bins as z-score of $\log_2$ IP/input. H3.3/H3.1 ratio, H3.3 and EdU 2 in S enrichment (z-score of $\log_2$ IP/input ratio of cpm) are shown at 10 kb bins smoothed over 3 non-zero bins. Two-tailed Mann-Whitney U test corrected for multiple testing by FDR (5% cut-off) was used to determine significance of differences between HIRA KO rescue with YFP (control) and HIRA or between HIRA KO + HIRA rescue signal in sites remaining in B or switching from B to A upon HIRA rescue. Significance was noted as: * ($p <= 0.05$), ** ($p <= 0.01$), *** ($p < = 0.001$) for all comparisons. $P$ values are for H3.3: KO + YFP vs HIRA at B-to-A $p = 0.02$, at B-to-B $p = 2.76e^{-19}$ and B-to-A vs B-to-B KO + HIRA $p = 0.56$, and for EdU:: KO + YFP vs HIRA at B-to-A $p = 4.76e^{-4}$, at B-to-B $p = 1.40e^{-21}$ and B-to-A vs B-to-B KO + HIRA $p = 0.05$.

---

This occurs specifically at non-transcribed sites without significant changes in expression. In contrast, buried sites in compartment A which were transcribed in WT cells remain there following HIRA deletion. Thus, loss of early initiation alone is not sufficient to result in a compartment change. Notably, the restoration in replication and H3.3 at buried sites did not require switching back from compartment B to A at our timescale upon HIRA rescue although this was sufficient to recover A compartment interactions. This implies that the role in HIRA in defining early IZs is independent of their compartment identity. To our knowledge, comparison of compartment organisation between unperturbed and impaired early firing IZs has not been reported to date. However, a global impairment of temporal replication control occurs upon depletion of RIF1[74,75] or MCM6[76]. Yet, only in the case of RIF1 changes in compartments have been identified in a cell type-specific manner[77,78]. However, it remained unclear how switching corresponded to changes in early initiation. In contrast, DNA methylation can impact compartment organisation, but there is conflicting evidence whether this is accompanied by changes in RT[79,80]. Here, we show that while HIRA plays a role in regulating compartment identity and compartment A interactions, this function is independent of its importance for early IZ definition.

In conclusion, our work demonstrates that HIRA regulates compartment A accessibility and 3D organisation independently of histone PTMs. In addition, HIRA defines early replication IZs independently of accessibility and histone H3 PTMs and irrespectively of the compartment they are in. We highlight how transcription-independent HIRA recruitment to early IZs provides a novel opportunity to understand how early replication and compartment organisation can be independently regulated.

## Methods

### Cell culture
We cultured HeLa cells stably expressing H3.1-SNAP-HA or H3.3-SNAP-HA that were either wild-type (WT) or HIRA knock out (KO) (CRISPR/Cas9-mediated)[50] and U2OS cells (ATCC HTB-96). Cell lines were grown in DMEM complete medium (Dulbecco's Modified Eagle's Medium with D-Glucose, L-Glutamine and Pyruvate) supplemented with 10% fetal calf serum, 100 U/mL Penicillin and 100 mg/mL Streptomycin. We routinely tested all cell lines to be free of mycoplasma.

### Cell transfection
We used siRNA to knock-down HIRA (acute HIRA depletion experiments) in H3.3-SNAP HeLa cells and U2OS cells. Briefly, $5 \times 10^5$ cells were seeded per 10 cm dish with a pre-mixed solution of Lipofectamine RNAiMAX (Thermo Fisher Scientific) and 15 µL 100 µM siRNA against HIRA or non-targeting control (Dharmacon), prepared according to the manufacturer's protocol. The siRNA–lipid complexes were prepared in Opti-MEM I Reduced Serum Medium and incubated

for 30 minutes at room temperature prior to cell addition, yielding a final siRNA concentration of 50 nM. Cells were transfected twice, at three-day intervals, over a total period of six days.

We performed rescue experiments in H3.1-SNAP-HA or H3.3-SNAP-HA HIRA KO HeLa cells by transfection with plasmids encoding HIRA-YFP or YFP as control[50] for 48 h using Lipofectamine 2000 (Thermo Fisher Scientific). For synchronised rescue experiments, we synchronized HeLa cells at the G1/S transition using a double thymidine block[19,81], starting the first block 6 h post-transfection (Fig. 5A). We monitored transfection efficiency 48 hours after transfection by detecting YFP by immunofluorescence microscopy.

### S phase entry detection by microscopy
Cell lines were synchronized at the G1/S transition using a double thymidine block and then released into S phase[19,81] for 2 h (HeLa) or 5 h (U2OS). To monitor replication, we performed EdU pulse (Supplementary Fig. 2, Supplementary Fig. 6E) or PCNA staining (Supplementary Fig. 6E). For EdU labelling, cells were pulsed with 20 µM EdU for 30 minutes[19]. EdU incorporation was detected using the Click-iT EdU Imaging Kit with Alexa Fluor 488 azide according to the manufacturer's instructions. For PCNA labelling, we performed pre-extraction and fixation as described above and then post-fixed cells with methanol for 20 minutes at −20 °C. Cells were blocked for with 5% BSA for 1 h prior to immunofluorescence staining performed[22] with PCNA antibody (DAKO, M879, 1:1000). Cells were co-stained with DAPI to label DNA. Images were acquired using a Zeiss Axiovert Z1 fluorescence microscope with 63× and 40× oil objective lenses, ORCA-Flash4.0 LT camera (Hamamatsu) and MetaMorph software. Image visualization and analysis was performed using ImageJ (Fiji) v1.54i.

### Hi-C
We performed Hi-C using the Arima Hi-C+ kit following the manufacturer's instructions. Briefly, about 5 million asynchronous HeLa cells per condition were fixed in 4% formaldehyde for 10 min before quenching the reaction with Stop Solution 1. Fixed cells were washed in PBS and snap-frozen in liquid nitrogen at 1 million cell aliquots. We performed cell and nuclear lysis, restriction enzyme digestion, end repair, biotinylation, ligation and decrosslinking as per manufacturer's instructions and proximally ligated DNA was isolated using AMPure XP beads. Arima QC1 was performed and was successful for all samples. We sonicated DNA using Covaris E220 Evolution (100 µL sample, 7 °C, peak incident power: 105 W, duty factor: 5%, cycles/burst: 200 for 100 s), and fragmented DNA was size selected using AMPure XP beads. We verified that the mean size of selected DNA was 400 bp using Tapestation. After biotin enrichment, we performed library preparation using the KAPA HyperPrep kit, following the modified protocol described in the Arima Hi-C+ kit instructions. After ligation of Illumina TruSeq sequencing adaptors, we performed Arima QC2 (library

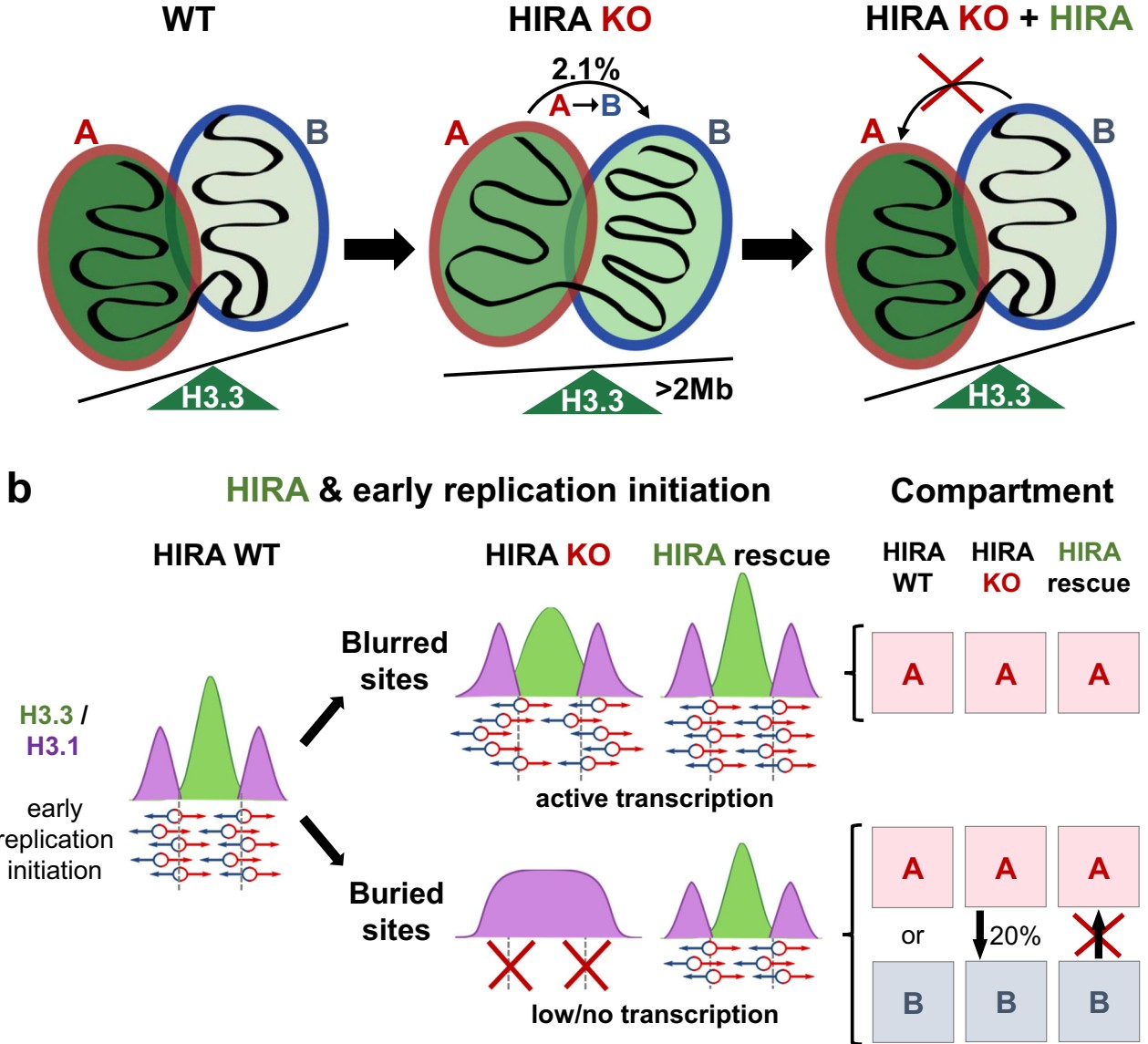

**Fig. 7 | HIRA defines early initiation zones independently of their 3D organization. a** In WT cells, HIRA promotes H3.3 deposition in compartment A, ensuring its enrichment in active chromatin. In the absence of HIRA, H3.3 is redistributed from compartment A to large (>2 Mb) B domains while compartment A accessibility increases. This is accompanied by minor compartment switching and reduction of compartment A interactions genome-wide without redistribution of H3 PTMs. Upon HIRA rescue, H3.3 targeting to compartment A and A-A interactions are restored without reversal of compartment switching. This indicates that HIRA plays a role in compartment organisation by regulating H3.3 deposition and nucleosome density, but independently from H3 PTM patterns. **b** HIRA-dependent early IZs (schematic adapted from Gatto et al., 2022[19] with permission from Elsevier) are predominantly found in compartment A when actively transcribed (blurred sites).

In contrast, early IZs can be found in both compartment A or B when they have low/no transcriptional activity (buried sites) in WT cells, indicating early initiation does not always occur in compartment A. Top: At blurred sites, H3.3 enrichment and early initiation become less precise in the absence of HIRA while remaining in compartment A. They regain sharpness upon rescue with HIRA also without changing compartment, indicating precision of initiation is not dependent on compartment identity. Bottom: Buried sites lose H3.3 enrichment and early firing in the absence of HIRA, but only a subset switches from compartment A to B. Furthermore, upon rescue with HIRA buried sites restore H3.3 enrichment and early firing without substantial switch from compartment B to A, indicating HIRA defined early replication initiation zones independently of their compartment organisation.

quantification) to determine the number of amplification cycles for the library PCR using the KAPA Library Quantification Sample kit following the manufacturer's instructions. Amplified libraries were sequenced on Illumina NovaSeq 6000 (PE100) at the NGS (Next-Generation Sequencing) platform at Institut Curie.

## Histone PTM ChIP-seq
We performed ChIP-seq of histone post-translational modifications (histone PTM ChIP-seq) using the native nucleosomes isolation procedure described in Gatto et al. (2022)[19] with small modifications. We used 5 million asynchronous cells per IP and Dynabeads Protein

A-conjugated antibodies against histone PTMs for immunoprecipitation. All steps were performed at 4 °C and in the presence of Protease inhibitors (Roche) and 1 mM TSA in every buffer to prevent protease and HDAC activity, respectively. For each IP reaction, we prepared 50 μL of Ab-conjugated beads by blocking for 4 h in Bead blocking buffer (2.5% BSA in PBS-T, 1 mg/mL tRNA), washing once in 0.02% PBS-T and incubating for 15–30 min with the antibodies (diluted in 0.2 mL 0.02% PBS-T) on a rotating wheel. Ab-conjugated beads were then washed twice with 0.02% PBS-T and resuspended in 0.2 mL 0.02% PBS-T. We pooled native nucleosomes (80 μL per IP) from up to 3 samples, diluted in 5x volumes of Incubation buffer (50 mM Tris-HCl pH 7.5, 100 mM NaCl, 0.5% BSA) and pre-cleared by incubating with Dynabeads Protein A (15 μL per input pool) for 30 min on a rotating wheel. We kept 20 μL (1%) pre-cleared chromatin as input sample. We incubated the remaining (460 μL/IP) with the Ab-conjugated beads (washed once in Incubation buffer) overnight on a rotating wheel and purified DNA as described for SNAP-seq[19]. We quantified and checked fragment size profile with Agilent 4200 TapeStation. Sequencing libraries were prepared at the NGS (Next-Generation Sequencing) platform at Institut Curie with the Illumina TruSeq ChIP kit and sequenced on Illumina NovaSeq 6000 (PE100).

## ATAC-seq

To assay accessibility, we performed ATAC-seq from 50000 G1/S-synchronised cells per condition using the Active Motif ATAC-seq kit (53150) following the manufacturer's instructions. We verified DNA profiles before sequencing on Illumina NovaSeq 6000 (PE100) by the NGS platform at Institut Curie.

## Total RNA-seq

We obtained total RNA from 1 million G1/S-synchronised cells per condition. We collected cells by trypsinisation and extracted RNA with the RNeasy Plus Mini Kit (QIAGEN) including DNase treatment (RNase-free DNase QIAGEN, 79254) using manufacturer's instructions. We quantified RNA using Nanodrop and checked the quality by Tapestation. We used 10 ng of total RNA for library preparation using TruSeq Stranded Total RNA kit and sequenced libraries on Illumina NovaSeq 6000 (PE100) at the NGS platform at Institut Curie.

## SNAP capture-seq and EdU-Seq

We performed SNAP capture-seq of transfected asynchronous or G1/S-synchronised cells by double thymidine block[19,81]. We carried out EdU labelling to map sites of ongoing synthesis from G1/S synchronized cells by a double thymidine block and released in S-phase[19]. Sequencing libraries were prepared at the Next Generation Sequencing (NGS) platform from Institut Curie with the Illumina TruSeq ChIP kit and sequenced on Illumina NovaSeq 6000 (PE100).

## Sequencing data processing

ChIP-seq and ATAC-seq data was processed from raw reads in FASTQ format as described in Gatto et al. (2022)[19]. Briefly, for each sample, we mapped reads to the soft-masked human reference genome (GRCh38) downloaded from Ensembl (release 109) using bowtie2 v2.3.4.2[82] with --very-sensitive parameters. RNA-seq data was aligned with hisat2 v2.1.0[83], run in paired-end mode with default parameters. We used SAMtools v1.9[84] to sort, flag duplicates and index bam files for all samples. We used samtools view (-f 2 -F 3840 parameters to keep reads mapped in pairs and exclude QC fails, non-primary alignments and duplicates) to compute coverage over the genome as a BED file (chromosome, start, end, MAPQ). Quality control of ATAC-seq data was additionally performed on bam files by ATACseqQC v1.28.0[85]. We used BEDtools v2.27.1[86] to calculate number of fragments in consecutive 100 bp or 1 kb bins.

Analysis of the data was carried out by custom Python scripts using pandas v1.5.3[87], NumPy v1.23.5[88] and scipy v1.11.2[89]. Visualization was performed using matplotlib v3.6.2[90] and seaborn v0.12.2[91]. For each sample, counts were read at 100 bp or at 1 kb resolution and aggregated to 10 kb bins. Binned data (at 100 bp and 10 kb) was normalized to the total number of mapped counts to generate cpm (counts per million). ChIP-seq data was then normalized to matched input sample by computing a $\log_2$ ratio (IP/input). To enable cross-sample comparison, ChIP-seq signal was then scaled and centered by computing a z-score per chromosome. RNA-seq data was plotted as $\log_2(cpm+1)$. Blurred ($n = 3274$) and buried ($n = 574$) site locations and H3.3 + OK-seq IZ ($n = 5596$) coordinates were obtained from Gatto et al. (2022)[19].

## ATAC-seq peak calling

ATAC-seq peaks were called from each replicate and condition using HMMRATAC[92] with default parameters except --window 2500000. Common peaks between the two replicates and cell lines for each condition were obtained by bioframe.overlap.

## Hi-C data processing

We used HiC-Pro v3.1.0[93] with default parameters (except MIN_MAPQ = 2) to generate raw count Hi-C matrices at 1 Mb, 100 kb, 50 kb, 25 kb, 10 kb and 5 kb resolution from raw FASTQ files. First, we used MultiQC v1.11[94] to perform quality control and extract the number of short-range (≤20 kb), long-range (>20 kb) cis and trans interactions for each sample. HiCExplorer v3.7.2[95–97] was then used to convert matrices to cool format and generate a single mcool file per sample, containing all resolutions listed above. Matrices were visualized interactively with HiGlass v0.8.0[98] at 1 Mb and 100 kb resolutions to be manually inspected for large inter- and intra-chromosomal aberrations (translocations, inversions, duplications, etc.), and the subsequently generated list of regions was merged with the set of blacklisted regions of the human genome (ensembl). This custom set of blacklisted regions was used to mask matrices prior to normalization by iterative correction (ICE)[99] with a single iteration using cooler v0.9.3[100]. Matrix similarity was computed per chromosome at 1 Mb, 100 kb, 50 kb and 10 kb resolution with HiCRep v0.2.6[101,102] before and after masking without substantial changes. Expected interactions per chromosome arm (coordinates downloaded with bioframe) were calculated using cooltools v0.6.1[103]. P(s) curves, representing decay of interaction frequency with increasing genomic distance, were calculated per chromosome arm at 10 kb resolution, aggregated and smoothed. Matrices from the H3.1- and H3.3-SNAP cell lines were analysed independently and showed similar results. Coordinates and data from H3.1-SNAP cells were used for representative images where experiments were performed in both cell lines unless mentioned otherwise.

## Hi-C data analysis

Compartment analysis of Hi-C matrices was performed by eigenvector (EV) decomposition[40] at 50 kb resolution with cooltools using GC content track to orient the sign of the first eigenvector (EV1). A/B compartment domains were defined for each sample as contiguous segments of the genome with the same EV1 sign. Compartment switching was determined on a per bin basis, where a compartment-switching bin undergoes the same EV1 sign change in H3.1-SNAP and H3.3-SNAP cell lines from HIRA WT to KO (WT-to-KO). As control, we compared compartment switching between H3.1-SNAP and H3.3-SNAP cells in the same way. Proportion of the genome switching compartment was calculated as the % compartment-switching 50 kb bins out of all non-masked 50 kb bins. Differential maps to visualize changes in compartment interactions were plotted as $\log_2$ ratio of ICE-normalized contacts from HIRA KO/WT at 1 Mb resolution. To assess A/B compartment interactions genome-wide, we performed saddle-plot analysis using cooltools. For this purpose, we split 50kb-binned EV1 signal into percentiles, re-ordered and averaged O/E-normalised Hi-C maps per chromosome arm from lowest (most strongly B) to highest

(most strongly A) EV1 percentile and plotted O/E interaction frequency at 50 × 50 bins. To generate differential saddle-plots, we calculated the $log_2$ ratio of HIRA KO/WT. TAD borders were identified based on insulation score[104], computed from 10 kb binned matrices with a window size of 100 kb using cooltools. Overlap between border bins (±20 kb) was determined using bioframe between the two cell lines and between the two conditions.

## Compartment analysis of blurred and buried sites

After masking the Hi-C data, we retained 2382 blurred and 439 buried sites that were used for compartment analysis. To assess their compartment identity, we re-binned the EV1 signal to 10 kb and we calculated the mean EV1 value from each sample. To determine compartment switches of the sites, we compared their compartment assignment between WT and HIRA KO (WT-to-KO) or HIRA KO + YFP to HIRA KO + HIRA rescue (YFP-to-HIRA). The proportion of sites changing or remaining in the same compartment are represented by the mean and standard deviation of the two cell lines. A set of random sites matched for size distribution was generated for comparison. As an additional control for the different compartment distribution of the blurred and buried sites, we also generated two sets of sites matching the size and compartment distribution of blurred and buried sites. As a reference for genome-wide behaviour in the scatterplots, we used 50kb-binned EV1 values.

## Quantifications and statistical analysis

Statistical analysis was performed in python using scipy[89]. *p*-values were calculated by two-tailed Mann-Whitney U test. Multiple testing correction was performed by controlling the false discovery rate (FDR) using the Benjamini-Hochberg method. Differences with adjusted p-value < 0.05 were considered statistically significant. Paired t-test or Welch's test (specified in the legends) was used to compare the differences in efficiency of transfection and S phase entry from cell counts. For each condition, >300 nuclei were quantified across two independent biological replicates. Quantification was performed manually and blinded to experimental condition.

## Reporting summary

Further information on research design is available in the Nature Portfolio Reporting Summary linked to this article.

## Data availability

The sequencing data generated in this study have been deposited on ArrayExpress with the following accession numbers: E-MTAB-14416 (H3 PTM ChIP-seq, https://www.ebi.ac.uk/biostudies/arrayexpress/studies/E-MTAB-14416), E-MTAB-14415 (ATAC-seq, https://www.ebi.ac.uk/biostudies/arrayexpress/studies/E-MTAB-14415), E-MTAB-14417 (RNA-seq, https://www.ebi.ac.uk/biostudies/arrayexpress/studies/E-MTAB-14417), E-MTAB-14433 (Hi-C, https://www.ebi.ac.uk/biostudies/arrayexpress/studies/E-MTAB-14433), E-MTAB-14419 (SNAP-seq and EdU-seq upon HIRA rescue, https://www.ebi.ac.uk/biostudies/arrayexpress/studies/E-MTAB-14419) and E-MTAB-14431 (Hi-C upon HIRA rescue, https://www.ebi.ac.uk/biostudies/arrayexpress/studies/E-MTAB-14431). Publicly available data H3.1- and H3.3- SNAP-seq at G1/S and EdU-seq at 2 h in S phase, as well as H3.3 site locations and H3.3-positive OK-seq IZs were obtained from Gatto et al. (2022)[19] (E-MTAB-10619, https://www.ebi.ac.uk/biostudies/arrayexpress/studies/E-MTAB-10619). Source data are provided with this paper. Details of all materials used are provided in Supplementary Table 3. Source data are provided with this paper.

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

## Acknowledgements

We thank the members of UMR3664 and Almouzni team for helpful discussions. We thank Sébastien Lemaire for critical reading of the Methods, Dominique Ray-Gallet for constructive feedback on the model, Héloïse Muller and Nicolas Servant for advice on the Hi-C experiments and analysis. We acknowledge the Cell and Tissue Imaging Platform PICT-IBiSA (member of France-Bioimaging ANR-10-INBS-04) of the UMR3664 and ICGex NGS platform of the Institut Curie. Funding to G.A. includes the European Research Council (ERC-2015-ADG-694694 'ChromADICT'), the Ligue Nationale contre le Cancer (Equipe labellisée Ligue), France and Agence Nationale de la Recherche, France (ANR-11-LABX-0044_DEEP, ANR-10-IDEX-0001-02 PSL, and ANR21-CE-11-0027 'CAFinDs'), Horizon EIC Pathfinder project 101099654 'RT-SuperES' and 'Cellular identities and destinies' exploratory research program (PEPR Cell-ID), a government grant managed by the Agence Nationale de la Recherche under the France 2030 program, with the reference number ANR-24-EXCI-0001, ANR-24-EXCI-0002, ANR-24-EXCI-0003, ANR-24-EXCI-0004, ANR-24-EXCI-0005'. T.K. was supported by H2020 MSCA-ITN—ChromDesign (Grant No. 813327) and La Ligue Nationale contre le Cancer (Grant No. TDLM23697). A.G. was supported by individual funding (H2020 MSCA-PF 'REPLICHROM4D', Grant No. 798106). R.N.V. is supported by Agence Nationale de la Recherche, France

(ANR-22-CE12-0018, 'IRONSUV'). M.A.M-R. acknowledges support by the Spanish Ministerio de Ciencia e Innovación (PID2020-115696RB-I00 and PID2023-151484NB-I00). L.M. is funded by the National Human Genome Research Institute (HG011536), the National Institute of General Medicine (GM 114190) and the National Science Foundation (Physics of Living Systems grant 1504942).

## Author contributions

G.A., J.P.Q., A.G. and T.K. conceived the overall strategy. T.K. and G.A. wrote the paper. T.K., A.F. and J.P.Q. performed the experiments. R.N.V. contributed to experiments in Supplementary Fig. 2. T.K. generated most of the figures and analysed data. G.A. supervised the work. M.A.M-R., A.G. and L.M. provided advice for the analysis. Critical reading and discussion of data involved all authors.

## Competing interests

The authors declare no competing interests.
