## [Transparent Peer Review file · Nature Communications]

HIRA defines early replication initiation zones independently of their genome compartment

Corresponding Author: Dr Geneviève Almouzni

Version 0:

Reviewer comments:

Reviewer #1

(Remarks to the Author)

This study examines the role of HIRA in regulating early replication initiation zones (IZs), demonstrating that HIRA's function is independent of chromatin accessibility, histone post-translational modifications (PTMs), and compartmental organization. The findings show that HIRA helps define early replication zones without relying on chromatin accessibility, as changes in accessibility at blurred and buried sites do not correlate with the loss of H3.3 enrichment or early replication. Additionally, the study reveals that HIRA does not influence early replication zones through histone PTMs, as active marks like H3K4me1, H3K4me3, and H3K27ac remain unchanged even without HIRA, indicating that histone marks alone cannot maintain early replication or H3.3 enrichment. Using Hi-C, the study further shows that blurred sites stay in compartment A regardless of HIRA status, while some buried sites shift from compartment A to B in HIRA knockout cells, although early replication at blurred sites is unaffected by transcription or compartment changes. HIRA complementation restores H3.3 enrichment, the balance between H3.3 and H3.1, and early replication initiation at both blurred and buried sites, but this does not depend on the reversal of compartment shifts. Interestingly, while blurred sites rely on transcription to recruit HIRA, buried sites recover H3.3 enrichment and early initiation independently of transcription, suggesting that DNA sequence features, residual HIRA complex subunits, or chromatin remodelers may serve as "memory" elements for HIRA recruitment at these sites. These results underscore HIRA's key role in defining early replication initiation zones, independent of chromatin accessibility, PTMs, or genome compartmentalization.

This paper makes a valuable contribution to our understanding of chromatin biology and replication by combining several advanced techniques, including ATAC-seq, ChIP-seq, EdU-seq, and Hi-C, to study how chromatin structure, HIRA function, and early replication zones (IZs) interact. The authors introduce a new framework for understanding "blurred" and "buried" IZs, which offers a better way to explore replication initiation in different chromatin environments. Their use of HIRA knockout and rescue models is a strong point, providing solid evidence for HIRA's role in shaping histone patterns and influencing early replication. A major strength of the paper is the finding that early IZs work independently of chromatin accessibility and histone modifications, challenging previous ideas and highlighting HIRA's unique regulatory role. The high-resolution data used in the study ensures the results are reliable, and the biological importance of the findings extends beyond basic chromatin biology, with wider implications for epigenetics. The paper is clearly written, with helpful visuals, making it both accessible and impactful. Additionally, the identification of potential ways HIRA is recruited opens up exciting possibilities for future research, with relevance to genome stability and diseases related to replication timing. Overall, the study is well-designed, offers new insights, and provides a strong basis for further exploration in the field.

Reviewer #2

(Remarks to the Author)

This manuscript by Almouzni and colleagues expands on the lab's previous work, which demonstrated that differential H3 variant targeting, controlled by HIRA, is important for the definition of early replication initiation zones (PMID: 35381196). HIRA loss-associated changes in H3.3 incorporation resulted in two types of early IZs: blurred sites where replication initiation became fuzzy at the boundaries and buried sites where H3.3 was lost along with abrogation of replication initiation. Here, the authors combine ATAC-seq, ChIP-seq for various histone marks, EdU-seq and Hi-C in the presence or absence of HIRA to further characterize and dissect these distinct early IZs. The data are well presented and their main conclusions are 1) HIRA plays a role in defining early replication timing zones independently of chromatin accessibility and H3 modification marks; 2) at some buried early RT zones, HIRA KO induces loss of early initiation that coincides with a compartment switch;

and 3) RT early initiation changes are independent of genome compartment changes. Of note, uncoupling of global RT from genome organization has been reported previously, as noted by the authors (PMID: 38081811). The finding that uncoupling of RT from compartmentalization can occur specifically in early RT zones presents only a modest advance from previous work in the absence of additional functional follow-up.

Specific criticisms:

- 1) All experiments are performed in HeLa cells. Validation of key observations in at least one additional cell line is needed.
- 2) It is interesting that HIRA re-expression can rescue the replication timing phenotype without affecting the compartment switch. The basis for this phenomenon should be explored in more detail. It is possible that the compartment switch is acquired as a result of prolonged replication timing defects and cannot be reversed through short-term HIRA re-expression. Have the authors considered stable re-introduction of HIRA over a prolonged time-period? Likewise, how does acute HIRA depletion affect RT and compartmentalization? The authors should further determine if the two processes may be attributable to distinct roles of HIRA, not all of which may relate to H3.3.
- 3) It would help to indicate in Fig. 3A, how EV1 values relate to A vs B compartments. In general, the term “A to B” switch is misleading as the data are more consistent with a model where B becomes more “B-like” in buried sites – see Fig 3B, negative EV1 median of buried sites in WT.

Reviewer #3

(Remarks to the Author)

Version 1:

Reviewer comments:

Reviewer #2

(Remarks to the Author)

The authors have adequately addressed our concerns through experimental additions, and by combining the two initial manuscripts into one compact paper that explores the impact of HIRA depletion on early replication initiation zones in the context of genome compartmentalization.

Reviewer #3

(Remarks to the Author)

Response to reviewers

We have considered reviewers' confidential feedback to merge our two initial manuscripts into one. For this significant reshaping, we have placed the focus on the replication part, since this part received the most positive feedback from both reviewers and provides a link to a physiological role of HIRA in the essential replication process.

Our revised version represents a single merged version of our submitted manuscripts and is entitled *HIRA defines early replication initiation zones independently of their genome compartment*, and the authors are T. Karagyoova, A. Gatto, A. Forest, J.-P. Quivy, R. Nunez-Vazquez*, M. Marti-Renom, L. Mirny & G. Almouzni ***Note that we added an additional author who contributed to the revision***

Given the focus on the second manuscript, we concentrated our efforts on addressing more specifically points raised by reviewers pertaining to the replication part. You will find below the list of changes in the figures.

Note that we have added additional experiments that are presented in the Extended Data: Extended Data Figure 2 is entirely new as well as Extended Data Figure 7D.

By assembling the two papers in a compact form and adding experiments and specific points to address reviewers comments we hope that this new and significantly improved version will now be satisfying for our reviewers.

Text changes:

In order to highlight the new text added and show how the two original manuscripts were merged, the manuscript text has been formatted such that new additions to the text are underlined and *sections coming from the original replication paper are in italics*.

Figure changes:

Main Figures:

1. Figure 1 combines components of Fig.1 and Fig. 4 from the 3D paper: Panels 1A-D are identical to those of the original 3D paper with the addition in Panel 1E of Fig. 4A from the original 3D paper
2. Figure 2 corresponds to Figure 3 from the original 3D paper
3. Figure 3 displays in Panel 3A and B the Fig.1A, C from the original replication paper, Panel 3C is Fig. 2A from the original replication paper
4. Figure 4 corresponds to Fig. 3 from the original replication paper
5. Figure 5 to Figure 4 from the original replication paper
6. Figure 6 to Figure 5 from the original replication paper
7. Figure 7 shows in Panel A an adapted schematic from Fig. 1E from the original 3D paper, Panel B reproduces Figure 6 from the original replication manuscript

Extended Data Figures:

1. Extended Data Figure 1: Identical with Extended Data Figure 1A-E from the original 3D organisation paper, except that the original Extended Data Figure 1D has been split into 1D & 1E (for H3.3 & H3.1, respectively), and thus the original Extended Data Figure E corresponds to the new Extended Data Figure F
2. Extended Data Figure 2: **new data**
3. Extended Data Figure 3: Extended Data Figure 1 from the original replication paper
4. Extended Data Figure 4: Extended Data Figure 4A is a merge of Figure 1B and Figure 2B from the original replication paper, Extended Data Figure 4B is Extended Data Figure 2B from the original replication paper
5. Extended Data Figure 5: Extended Data Figure 3A-C from the original replication paper
6. Extended Data Figure 6: Extended Data Figure 5 from the original replication paper
7. Extended Data Figure 7: Extended Data Figure 7A-C, E are Extended Data Figure 6A-D from the original replication manuscript with **new data** in Extended Data Figure 7D

Point-by-point response to comments raised to the second ‘replication manuscript’

Reviewer #1, replication manuscript (Remarks to the Author)

This study examines the role of HIRA in regulating early replication initiation zones (IZs), demonstrating that HIRA’s function is independent of chromatin accessibility, histone post-translational modifications (PTMs), and compartmental organization. The findings show that HIRA helps define early replication zones without relying on chromatin accessibility, as changes in accessibility at blurred and buried sites do not correlate with the loss of H3.3 enrichment or early replication. Additionally, the study reveals that HIRA does not influence early replication zones through histone PTMs, as active marks like H3K4me1, H3K4me3, and H3K27ac remain unchanged even without HIRA, indicating that histone marks alone cannot maintain early replication or H3.3 enrichment. Using Hi-C, the study further shows that blurred sites stay in compartment A regardless of HIRA status, while some buried sites shift from compartment A to B in HIRA knockout cells, although early replication at blurred sites is unaffected by transcription or compartment changes. HIRA complementation restores H3.3 enrichment, the balance between H3.3 and H3.1, and early replication initiation at both blurred and buried sites, but this does not depend on the reversal of compartment shifts. Interestingly, while blurred sites rely on transcription to recruit HIRA, buried sites recover H3.3 enrichment and early initiation independently of transcription, suggesting that DNA sequence features, residual HIRA complex subunits, or chromatin remodelers may serve as “memory” elements for HIRA recruitment at these sites. These results underscore HIRA’s key role in defining early replication initiation zones, independent of chromatin accessibility, PTMs, or genome compartmentalization.

This paper makes a valuable contribution to our understanding of chromatin biology and replication by combining several advanced techniques, including ATAC-seq, ChIP-seq, EdU-seq, and Hi-C, to study how chromatin structure, HIRA function, and early replication zones (IZs) interact. The authors introduce a new framework for understanding “blurred” and “buried” IZs, which offers a better way to explore replication initiation in different chromatin environments. Their use of HIRA knockout and rescue models is a strong point, providing solid evidence for HIRA’s role in shaping histone patterns and influencing early replication. A major strength of the paper is the finding that early IZs work independently of chromatin accessibility and histone modifications, challenging previous ideas and highlighting HIRA’s unique regulatory role. The high-resolution data used in the study ensures the results are reliable, and the biological importance of the findings extends beyond basic chromatin biology, with wider implications for epigenetics. The paper is clearly written, with helpful visuals, making it both accessible and impactful. Additionally, the identification of potential ways HIRA is recruited opens up exciting possibilities for future research, with relevance to genome stability and diseases related to replication timing. Overall, the study is well-designed, offers new insights, and provides a strong basis for further exploration in the field.

We thank this reviewer who is positive about the work and does not ask for specific novel experiments to perform. Considering the suggestion to make a single paper in the confidential comments, we hope that our revision scheme in which we propose to include parts from our 3D paper will further strengthen the story and thus should satisfy both reviewers. The comments provided on the two manuscripts proved very useful. Given the encouraging evaluation of the importance of our insights into the role of HIRA in defining early replication initiation zones. Following the comments from both reviewers, we merged the two manuscripts into one mainly focused on the replication aspect.

Reviewer #2, replication manuscript (Remarks to the Author)

This manuscript by Almouzni and colleagues expands on the lab's previous work, which demonstrated that differential H3 variant targeting, controlled by HIRA, is important for the definition of early replication initiation zones (PMID: 35381196). HIRA loss-associated changes in H3.3 incorporation resulted in two types of early IZs: blurred sites where replication initiation became fuzzy at the boundaries and buried sites where H3.3 was lost along with abrogation of replication initiation. Here, the authors combine ATAC-seq, ChIP-seq for various histone marks, EdU-seq and Hi-C in the presence or absence of HIRA to further characterize and dissect these distinct early IZs. The data are well presented and their main conclusions are 1) HIRA plays a role in defining early replication timing zones independently of chromatin accessibility and H3 modification marks; 2) at some buried early RT zones, HIRA KO induces loss of early initiation that coincides with a compartment switch; and 3) RT early initiation changes are independent of genome compartment changes. Of note, uncoupling of global RT from genome organization has been reported previously, as noted by the authors (PMID: 38081811). The finding that uncoupling of RT from compartmentalization can occur specifically in early RT zones presents only a modest advance from previous work in the absence of additional functional follow-up.

We thank this reviewer for the summary of our work. However, we should clarify the fact that we are focusing on early replication IZs and not early RT in general. Additionally, it is important to note that so far, the uncoupling of RT from compartments has been shown only in the context of early development (Dileep et al., 2019; Miura et al., 2019; Nakatani et al., 2023), where neither genome organisation nor RT are yet definitively set up. The situation in RIF1 KO in human cells (Klein et al., 2021), is very distinct because it gives rise to increased cell-to-cell variability in RT accompanied by compartment switches determined by population-based Hi-C, without a clear correspondence between the two. We therefore believe that our finding here with the uncoupling that we discovered in a cellular system where 3D organization is already established is novel. Furthermore, the unique role of HIRA in this context, with a role in 3D organization distinct from the one on early replication IZ is an important and entirely new contribution. As stated by the other reviewer, this can open exciting possibilities for future research.

Specific criticisms:

1) All experiments are performed in HeLa cells. Validation of key observations in at least one additional cell line is needed.

To address this point which allows to broaden the impact of our findings, we have performed acute HIRA depletion by siRNA in HeLa and U2OS cells. We find that it recapitulates the delayed S phase entry phenotype observed in the HIRA KO cells (Extended Data Figure 2). This data, in combination with the rescue of the constitutive HIRA KO cells by 48h transient transfection of HIRA indicates that the phenotype we observe in the HIRA KO cells cannot reflect an adaptation to the loss of HIRA. We have thus addressed the comment of this reviewer about the original 3D organization manuscript.

2) It is interesting that HIRA re-expression can rescue the replication timing phenotype without affecting the compartment switch. The basis for this phenomenon should be explored in more detail. It is possible that the compartment switch is acquired as a result of prolonged replication timing defects and cannot be reversed through short-term HIRA re-expression. Have the authors

considered stable re-introduction of HIRA over a prolonged time-period? Likewise, how does acute HIRA depletion affect RT and compartmentalization?

We appreciate this reviewer's point and agree that the absence of compartment switching for buried sites 48h post-rescue does not necessarily mean that it may not appear at a later timepoint. However, we note a recovery in the global defect in A-A compartment interactions for the 48h HIRA rescue which we now show as a new piece of information (Extended Data Figure 7D, also addressing a comment from the reviewer about the original 3D organization manuscript). This observation implies that there is indeed global recovery of compartment contacts without necessarily a switching for the buried sites. Furthermore, the novel point of the current study was to demonstrate that compartment switching is not necessary for recovery of early replication initiation at buried sites.

At this point in time, we have not explored further stable and (long-term) re-introduction of HIRA, since we focused on the early disconnection from replication recovery. However, to go further into the 3D aspects, how prolonged exposure to HIRA could allow compartment switch as raised by the reviewer would be very interesting.

The authors should further determine if the two processes may be attributable to distinct roles of HIRA, not all of which may relate to H3.3.

Whether it is HIRA's role in H3.3 deposition or an independent function which is important for early replication initiation and compartment organisation is an interesting point. We discussed this aspect and carefully edited the manuscript to stress that it is HIRA-dependent. Indeed, previous work has put forward functions of HIRA independent of H3.3 in the context of DNA repair (Bouvier et al., 2021) and we now quote this work. Future studies will need an extensive analysis to explore this aspect in depth to be able to draw proper conclusions and we believe that this may form on its own an important story which is beyond the scope of the current manuscript especially now that we have merged our two stories. Nevertheless, our data indicate that H3.3 distribution both with respect to compartments and early IZs is restored upon acute re-expression of HIRA in the HIRA KO cells, indicating that H3.3 deposition is likely connected to the rescue of both the A-A compartment interactions and the replication defects. Accordingly, to avoid any misunderstanding, in line with this reviewer's comment, we have now carefully reworded the text and discussion to place the emphasis on HIRA.

3) It would help to indicate in Fig. 3A, how EV1 values relate to A vs B compartments. In general, the term "A to B" switch is misleading as the data are more consistent with a model where B becomes more "B-like" in buried sites – see Fig 3B, negative EV1 median of buried sites in WT.

We thank the reviewer for pointing this out. On the scatterplot in Figure 4A (old Figure 3A), we have shown the EV1 values and have now adapted the figure to include their correspondence to A/B compartments. Furthermore, we have described that the EV1 values shift down for buried sites as a whole (Figure 4B, old Figure 3B), but not all of them are in compartment B in WT cells – 40% buried sites are actually in compartment A in WT cells and only half of them experience a strong enough EV1 value reduction to switch to compartment B (Figure 4A, old Figure 3A).

Response to reviewers after revision

REVIEWERS' COMMENTS

Reviewer #2 (Remarks to the Author):

The authors have adequately addressed our concerns through experimental additions, and by combining the two initial manuscripts into one compact paper that explores the impact of HIRA depletion on early replication initiation zones in the context of genome compartmentalization.

Reviewer #3 (Remarks to the Author):

We thank the reviewers for their positive evaluation of our revised manuscript